# V-STaR: Training Verifiers for Self-Taught Reasoners

**Arian Hosseini** ❋ *    **Xingdi Yuan** 🔲    **Nikolay Malkin** ❋⊛
**Aaron Courville** ❋    **Alessandro Sordoni** ❋🔲    **Rishabh Agarwal** ◉

❋Mila, Université de Montréal
🔲Microsoft Research
⊛University of Edinburgh
◉Google Deepmind

## Abstract

Common self-improvement approaches for large language models (LLMs), such as STaR (Zelikman et al., 2022), iteratively fine-tune LLMs on self-generated solutions to improve their problem-solving ability. However, these approaches discard the large amounts of incorrect solutions generated during this process, potentially neglecting valuable information in such solutions. To address this shortcoming, we propose V-STaR that utilizes both the correct and incorrect solutions generated during the self-improvement process to train a verifier using DPO that judges correctness of model-generated solutions. This verifier is used at inference time to select one solution among many candidate solutions. Running V-STaR for multiple iterations results in progressively better reasoners and verifiers, delivering a 4% to 17% test accuracy improvement over existing self-improvement and verification approaches on common code generation and math reasoning benchmarks with LLaMA2 models.

## 1 Introduction

Learning to recognize and correct mistakes is a feature of human intelligence (Metcalfe, 2017). When dealing with complex tasks, such as coding or solving a math problem, we can recognize errors in reasoning and explore alternative paths to a solution. To improve the reasoning performance of LLMs, several approaches exploit the ability of LLMs to produce solutions and check the correctness of these solutions during training, for example, using test cases for code generation. These self-improvement approaches, such as STaR Zelikman et al. (2022), RFT (Yuan et al., 2023), and ReST$^{EM}$ (Singh et al., 2023), improve LLMs by fine-tuning them on their self-generated solutions and optionally iteratively running this process. However, all these approaches are data-inefficient in that they use *only* correct solutions, and discard incorrect solutions, which is often a large portion of model-generated solutions, especially for challenging reasoning tasks.

Orthogonal to self-improvement, another promising direction to improve LLM reasoning is to use learned LLM verifiers at test time (Cobbe et al., 2021; Wang et al., 2023b). Specifically, the LLM generates multiple candidate solutions and the verifier ranks these solutions and selects the best one. Such verifiers are trained by fine-tuning an LLM on a dataset of solutions generated from a frozen LLM, labeled with either final correctness (ORM, Cobbe et al., 2021) or step-by-step human annotations (Lightman et al., 2024). Using such verifiers allows LLMs to trade-off additional test-time compute for better performance.

We propose **V**erification for **S**elf-**Ta**ught **R**easoners (V-STaR). The key idea in V-STaR is to utilize both the correct and incorrect LLM-generated solutions during the iterative self-improvement process to train a verifier using DPO[1] (Rafailov et al., 2023), in addition to training a LLM as generator using correct solutions (Fig. 1). The iterative self-improvement process yields progressively improved generators, trained on augmented data, which leads

---

*Correspondence to: <arian.hosseini9@gmail.com> and <rishabhagarwal@google.com>
[1]We also considered using this DPO setup to train a generator in §4.6.

Figure 1: **Generator and verifier training in V-STaR**. **Left:** In each training iteration, the generator $G^t$ is fine-tuned (from a pretrained LLM) on the current buffer of problem instances and correct solutions $\mathcal{D}_{\text{GEN}}$. Generated solutions that yielded a correct answer are added to $\mathcal{D}_{\text{GEN}}$ to be used in future iterations, and all the generated solutions (correct and incorrect) are added to $\mathcal{D}_{\text{VER}}$. The verifier $V^t$ is trained using DPO with a preference dataset constructed from pairs of correct and incorrect solutions from $\mathcal{D}_{\text{VER}}$. **Right:** At test time, the verifier is used to rank solutions produced by the generator. Such iterative training and inference-time ranking yields large improvements over generator-only self-improvement.

to higher quality completions and more challenging negative examples for the verifier training. At test time, the verifier ranks multiple candidate solutions from the generator and selects the best one.

We empirically evaluate V-STaR to improve reasoning capabilities of LLMs: (1) Math problem-solving using GSM8K (Cobbe et al., 2021) and a subset of MATH (Hendrycks et al., 2021), and (2) Code-generation using MBPP (Austin et al., 2021) and HumanEval (Chen et al., 2021). Fine-tuning LLaMA2 (Touvron et al., 2023) and CodeLLaMA (Rozière et al., 2023), we compare V-STaR to other self-improvement (RFT, STaR) and verification-based methods (ORM), self-consistency (Wang et al., 2023a), and a non-iterative V-STaR baseline (RFT + Verifier) that uses the same number of generation samples to bootstrap a generator and verifier. V-STaR works remarkably well, leading to 6% to 17% absolute improvement in test accuracy over prior self-improvement and verification-based methods for math reasoning, and 4% to 12% in code generation. Notably, 7B V-STaR surpasses base LLaMA2 70B (8-shot) on GSM8K, and nearly match CodeLLaMA 34B (zero-shot) on HumanEval.

Our contributions are:

- We propose V-STaR, a simple and effective approach, which uses iteratively generated correct and incorrect solutions from an LLM to train a better generator and verifier. V-STaR outperforms prior self-improvement approaches (RFT, STaR) as well as ORM verification (Cobbe et al., 2021) for math reasoning and code generation (Fig. 2). V-STaR better utilizes adaptive test-time compute for improving performance than strong baselines – ORM verification, RFT + verifier (Fig. 5), and self-consistency (Fig. 6, left).
- As a secondary contribution, we find DPO to be more effective for training verifiers than the prevalent ORM approach by Cobbe et al. (2021). We also propose a formula for Best-of-$k$ (§4.2), akin to Pass@$k$, to reliably evaluate test performance with verification.

## 2 Preliminaries

Given a pretrained language model $G$ and the original training data of a task $\mathcal{D}_{\text{SFT}} = \{(x_1, y_1), (x_2, y_2), \cdots (x_N, y_N)\}$, where $x$ is typically a description of a problem and $y$ is the solution, such as chain-of-thought rationale or generated code. The de facto approach for such tasks with causal language models is supervised fine-tuning (SFT) with the negative log-likelihood objective on the training data:

$$\mathcal{L}_{\text{SFT}}(G) = -\mathbb{E}_{(x,y) \sim \mathcal{D}_{\text{SFT}}} \sum_{t=1}^{T} \log G(y_t \mid y_{<t}, x) \tag{1}$$

where $G$ is also referred to as *generator* in reasoning tasks. LLMs can be used to generate high quality chain-of-thought rationales or solutions for a range of tasks. This observation

Table 1: Comparison of self-improvement and verification methods, showing the data used to train the generator and verifier (if applicable), and whether or not the method is iterative.

| METHOD | GENERATOR DATA | VERIFIER DATA | ITERATIVE |
|---|---|---|---|
| SFT | $\mathcal{D}_{\text{SFT}}$ | ✗ | ✗ |
| VERIFICATION | $\mathcal{D}_{\text{SFT}}$ | $\mathcal{D}_{\text{SFT}} \cup$ GENERATED | ✗ |
| STAR | CORRECT GENERATED$_{t-1}$ | ✗ | ✓ |
| RFT (STAR$^{\dagger}$[1 ITER]) | $\mathcal{D}_{\text{SFT}} \cup$ CORRECT GENERATED | ✗ | ✗ |
| STAR$^{\dagger}$ | $\mathcal{D}_{\text{SFT}} \cup$ CORRECT GENERATED$_{<t}$ | ✗ | ✓ |
| V-STAR [1 ITER] | $\mathcal{D}_{\text{SFT}} \cup$ CORRECT GENERATED | $\mathcal{D}_{\text{SFT}} \cup$ GENERATED | ✗ |
| **V-STAR** | $\mathbf{\mathcal{D}_{\text{SFT}} \cup}$ **CORRECT GENERATED**$_{<t}$ | $\mathbf{\mathcal{D}_{\text{SFT}} \cup}$ **GENERATED**$_{<t}$ | ✓ |

has motivated using correct generations from the model itself to bootstrap problem-solving and reasoning abilities (Zelikman et al., 2022; Singh et al., 2023; Yuan et al., 2023).

## 2.1 Self-improvement approaches

**Self-Taught Reasoner** (STaR; Zelikman et al., 2022) corresponds to an iterative approach where a language model improves itself using correctness feedback. In each iteration, one solution $\hat{y}$ is generated using greedy decoding from the generator $G$ for each problem $x$ in training dataset $\mathcal{D}$. Having access to test cases or ground truth answers, generated solutions can be checked for their binary correctness label $z$ by $z = \text{is\_correct}(x, \hat{y})$, $z \in \{0, 1\}$. A completion $\hat{y}$ is labeled correct if it has the same final answer as the ground truth answer for math problems, or if it passes all the test cases for code generation problems. Only correct solutions ($z = 1$) are included in the dataset at iteration $j$ where $\mathcal{D}_j = \{(x_1, \hat{y}_1), (x_2, \hat{y}_2), \cdots (x_N, \hat{y}_N)\}$. Then, the generator is fine-tuned on this new dataset using (Eq. 1) where $\mathcal{D}_{\text{SFT}} = \mathcal{D}_j$. This fine-tuned generator is used in subsequent iterations.

**Rejection Sampling Fine-tuning** (RFT; Yuan et al., 2023) first fine-tunes a pretrained LM on the training dataset $\mathcal{D}_{\text{SFT}}$ to obtain $G$. For each problem $x_i$, $k$ solutions are sampled $\{\hat{y}_{i,j} \sim G(y|x_i)\}_{j=1}^{k}$ and similar to STaR, only correct generated solutions ($z = 1$) are kept. In RFT, the original dataset is then augmented with the correct completions to $\mathcal{D}_j$, and $G$ is fine-tuned on the new $\mathcal{D}_j$ to obtain $G_{\text{RFT}}$. Unlike STaR, RFT is not an iterative approach.

**STaR$^{\dagger}$**. Each STaR iteration can be performed similarly to RFT, akin to ReST$^{EM}$ (Singh et al., 2023). Since there could be multiple correct solutions for a problem, one could sample $k$ solutions per problem at each STaR iteration, but this is not prescribed in the original STaR paper, which we will denote as STaR$^{\dagger}$ for the rest of the paper.

## 2.2 Test-time verification

Cobbe et al. (2021) trained verifiers, which they refer as outcome-supervised reward model (**ORM**), that assess the probability that a candidate solution is correct for a given problem. At test time, the language model $G$ generates many candidate solutions and the one ranked highest by the verifier is selected, also known as **Best-of-k**. To train a verifier $V$, similar to RFT, $k$ candidate solutions are sampled from a generator $G$ for each training problem and labeled for their correctness $z$ to make the verifier training data $\mathcal{D}_{\text{VER}} = \{(x_i, \hat{y}_{i,j}, z_{i,j})\}_{i=1}^{N}$, where $z_{i,j}$ is a binary label indicating whether $\hat{y}_{i,j}$ is a correct or incorrect solution.

To train the verifier $V$, Cobbe et al. (2021) fine-tune a LLM on $\mathcal{D}_{\text{VER}}$ using a combination of language modeling (Eq. 1) and binary classification. The model is trained to predict $\hat{y}_{i,j}$ given $x_i$ and $z_{i,j}$ given $\{x_i; \hat{y}_{i,j}\}$ with the language modeling and the classification objective, respectively. See §5 for more details.

## 2.3 Preference learning with DPO

Fine-tuning pretrained LLMs with human feedback can result in large performance gains in downstream tasks Ouyang et al. (2022); Bai et al. (2022). The typical framework to do so is to

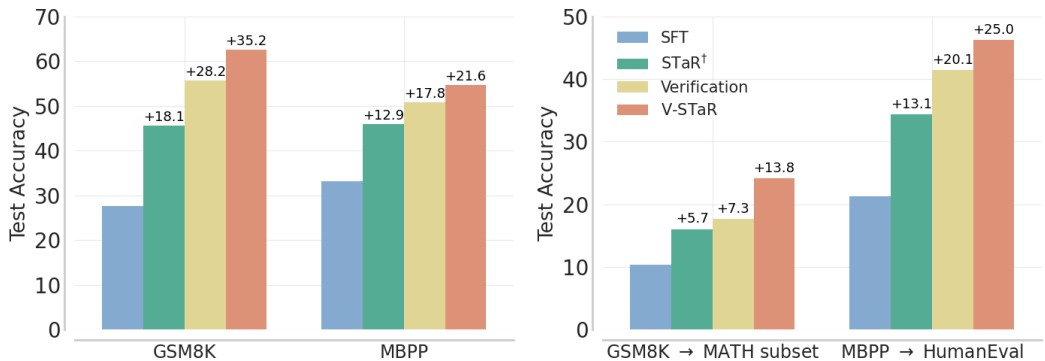

Figure 2: **Test accuracy of 7B V-STaR compared to self-improvement and verification baselines.** We report Best-of-64 for verification-based methods and Pass@1 for others. All methods, except SFT, have access to the SFT baseline model and $K = 48$ output generations per problem. STaR$^\dagger$ and V-STaR are run for 3 iterations, where each iteration uses $K/3 = 16$ samples. Verification corresponds to using a test-time verifier trained on K generated completions from the SFT generator. Numbers above each bar show the absolute improvement over SFT. **(Left)** Test accuracy on tasks used for V-STaR training. **(Right)** Transfer evaluation of GSM8K and MBPP trained models on MATH subset and HumanEval respectively.

collect paired human preferences for a set of input prompts $\mathcal{D}_{\text{pref}} = \{(x_i, y_i^+, y_i^-)\}_{i=1}^N$, train a reward model using $\mathcal{D}_{\text{pref}}$, and fine-tune the LLM using this reward (Stiennon et al., 2020).

More recently, Rafailov et al. (2023) proposed Direct Preference Optimization (DPO), which does not use a separately trained reward model during fine-tuning. DPO requires supervised fine-tuning (SFT) a pretrained LLM on the downstream task to obtain $G_{\text{SFT}}$, which is also used as a reference policy. Given the preference dataset $\mathcal{D}_{\text{pref}}$ and $G_{\text{SFT}}$, DPO's objective increases the log likelihood, relative to the reference policy, of preferred $y^+$ to dispreferred $y^-$ completions. We present the DPO loss later in Eq. 2, in the context of training verifiers.

## 3 V-STaR: Verifiers for self-taught reasoners

Existing self-improvement methods, such as RFT, STaR, and ReST$^{EM}$, throw away incorrect model-generated solutions. However, incorrect solutions can also contain valuable information: a language model could learn from discrepancies between correct and incorrect solutions for a given problem, and identify error patterns in generations, enhancing its ability to provide more accurate solutions. In this work, we propose **V-STaR** that utilizes both incorrect and correct generated solutions in an iterative process to train a better generator and verifier (see algorithm 1 in appendix).

- First, we fine-tune a pretrained LLM $G_{\text{base}}$ on the original training data $\mathcal{D}_{\text{SFT}}$ to obtain generator $G_{\text{SFT}}$.

- Next, we sample $k$ completions for each problem in the training data from the generator $\{\hat{y}_{i,j} \sim G(y|x_i)\}_{j=1}^k$, where $x \in \mathcal{D}_{\text{query}}$ (see §D for an example).

- Generated solutions are labeled for their correctness $z$ using ground truth answers or test cases. We use only correct generated solutions ($z = 1$) to augment the generator training data $\mathcal{D}_{\text{GEN}}$ as $(x_i, \hat{y}_{i,j})$. Both correct and incorrect generated solutions are added to verifier data $\mathcal{D}_{\text{VER}}$ with their correctness label as $(x_i, \hat{y}_{i,j}, z_{i,j})$, so the verifier can learn from generator's mistakes.

- In the next iteration $t$, the generator $G^t$ is obtained by fine-tuning the pretrained model $G_{\text{base}}$ on the augmented $\mathcal{D}_{\text{GEN}}$. We can sample solutions again from this generator $G^t$. This process is repeated for up to $T$ iterations to augment $\mathcal{D}_{\text{GEN}}$ and $\mathcal{D}_{\text{VER}}$ iteratively.

- The final generator $G^T$ is obtained by using $\mathcal{D}_{\text{GEN}}$ to fine-tune a pretrained model $G_{\text{base}}$. The verifier $V^T$ is obtained by using $\mathcal{D}_{\text{VER}}$ to further train a model $G_{\text{SFT}}$ which was fine-tuned on the original $\mathcal{D}_{\text{SFT}}$.

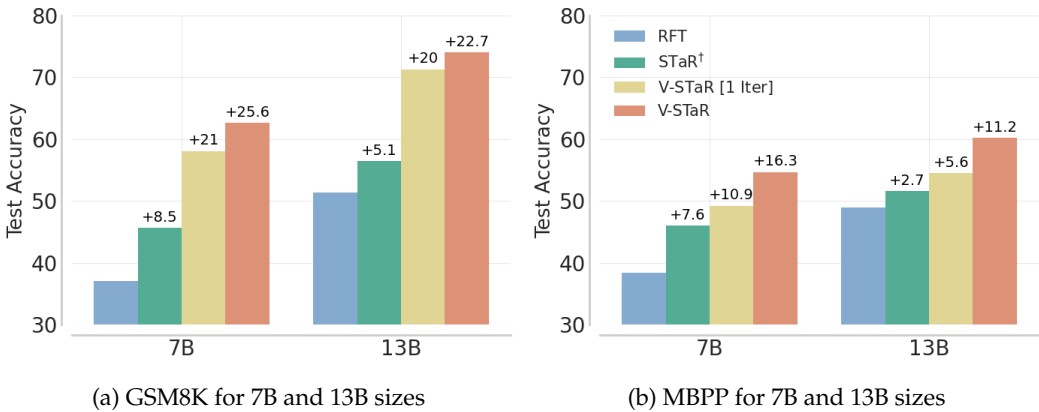

(a) GSM8K for 7B and 13B sizes  (b) MBPP for 7B and 13B sizes

Figure 3: **Pass@1 and Best-of-64 scores for generator-only and verifier-based methods**. Numbers above each bar represent the absolute improvement over RFT. V-STaR[1 Iter] baseline is trained with data consisting of $3 \times 16$ completions per query for one iteration only. STaR[†]and V-STaR are trained using iterative data collection (i.e. 16 completions generated per query at each iteration). At test time, Best-of-64 is calculated using Eq. 3 on 128 candidate answers sampled per problem from the generators. V-STaR 7B performs on par with CodeLLaMA 34B, which has a zero-shot Pass@1 of 55% on MBPP.

In our approach, the original training data is also included as correct solutions in both generator data and verifier data. The difference between V-STaR training and Cobbe et al. (2021) is that our verifier training data is collected iteratively, each iteration from a better generator, while ORM only collects data from a fixed generator that is only fine-tuned on the original SFT data. We compare to this ORM approach as a baseline, as discussed in §4.1.

### 3.1 Training verifiers with DPO

Following Cobbe et al. (2021), current LLM verifiers are trained with a combination of language modeling and binary classification loss (§2.2). These two objectives can be unified via offline preference learning methods, such as DPO (Rafailov et al., 2023), where the proximity to the reference policy is a proxy for the language modeling objective while the classification loss is a proxy for reward modelling. Empirically, we found DPO verifiers to be better than ORM-style verifiers (§4.4), when using LoRA adapters (Hu et al., 2022).

To use DPO for training verifiers, we construct a preference pair dataset using collected solutions in $\mathcal{D}_{\text{VER}}$. We treat correct solutions as preferred and incorrect solutions as not preferred completions given the problem. Specifically, $\mathcal{D}_{\text{VER}} = \{(x_i, y_{i,1}^+, y_{i,1}^-), \cdots, (x_i, y_{i,m}^+, y_{i,m}^-)\}_{i=1}^N$, where $m$ is the number of preference pairs which are from the Cartesian product of correct and incorrect solutions. We train our verifier $V$ using this constructed $\mathcal{D}_{\text{VER}}$ and the SFT policy $G_{\text{SFT}}$ using the DPO objective, $\mathcal{L}_{\text{DPO}}(V; G_{\text{SFT}})$:

$$-\mathbb{E}_{(x,y^+,y^-)\sim\mathcal{D}_{\text{VER}}} \left[\log \sigma \left(\hat{r}(x, y^+) - \hat{r}(x, y^-)\right)\right], \quad \text{with } \hat{r}(x,y) = \beta \log \frac{V(y|x)}{G_{\text{SFT}}(y|x)} \quad (2)$$

where $\sigma$ is the logistic function, and $\beta$ is a hyper-parameter controlling the proximity to the reference policy $G_{\text{SFT}}$. The DPO objective steers the verifier towards increasing the likelihood of correct solutions $y^+$ and decreasing the likelihood of incorrect solutions $y^-$ for a problem $x$. At inference, we use the likelihood of a generated solution given a problem under the trained DPO verifier (i.e. $V(\hat{y}|x)$) as scores to rank candidate solutions.

## 4 Empirical results

To demonstrate the effectiveness of V-STaR, we conduct experiments on two widely used datasets: GSM8K (Cobbe et al., 2021) for solving math problems, and MBPP (Austin et al., 2021) for code-generation problems. We also evaluate the transfer generalization performance of V-STaR using Hendrycks' MATH (Hendrycks et al., 2021) and HumanEval (Chen et al., 2021). Specifically, for math reasoning, we only train our generators and verifiers

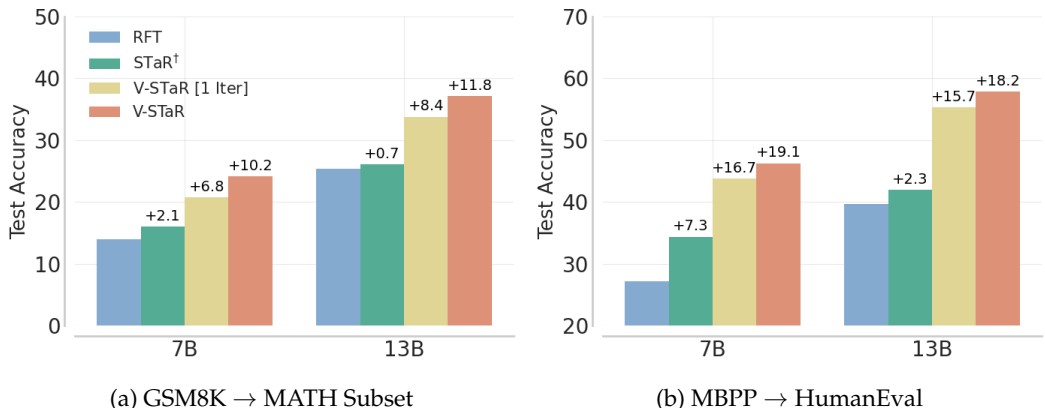

(a) GSM8K → MATH Subset        (b) MBPP → HumanEval

Figure 4: **Out-of-domain transfer evaluation**: Pass@1 and Best-of-64 for generators and verifiers with absolute improvement over RFT shown above each bar. Models trained on GSM8K are evaluated on a subset of MATH test set (§4), and models trained on MBPP are evaluate on HumanEval test set. V-STaR 7B performs close to CodeLLaMA 34B which has a zero-shot Pass@1 of 48% on HumanEval.

using GSM8K training data and evaluate them on the whole GSM8K test set and a subset of MATH test set[2]. For code generation, we train our models using the MBPP training data and evaluate them on the full test sets of MBPP and HumanEval.

**Models**    For experiments, we fine-tune LLaMA2 (Touvron et al., 2023) and CodeLLaMA (Rozière et al., 2023) 7B and 13B models using LoRA (Hu et al., 2022). Generators are trained with a causal language modeling objective, and our baseline (V-STaR[1 Iter]) and V-STaR verifiers are trained using DPO. The reference policy $G_{\text{SFT}}$ for DPO is trained on the original training data for 2 and 3 epochs for GSM8K and MBPP, respectively. See §3.1 for details.

**Data generation**    For each iteration, $k = 16$ completions are sampled per query from the previous iteration's generator. For GSM8K, the first iteration samples are from a generator trained solely on the original GSM8K training data for 2 epochs. For MBPP, this data is from a 3-shot pretrained CodeLLaMA (see §B). Completions are labeled for correctness by checking the final answer for math problems and running test cases for coding problems.

## 4.1 Baselines and metrics

We run V-STaR for 3 iterations and sample $k = 16$ solutions at each iteration to augment $\mathcal{D}_{\text{GEN}}$ and $\mathcal{D}_{\text{VER}}$. To assess the gains from our iterative approach, we compare against a number of baselines (Table 1):

1. **SFT**: Standard fine-tuning (Eq. 1) on training data without any self-improvement or test-time verification.
2. **STaR[†]**: Bootstrapping a generator by sampling $k = 16$ completions per query for 3 iterations, see §2.1.
3. **RFT**: Running STaR[†] by sampling $3 \times 16$ completions for only 1 iteration, see §2.1.
4. **Verification** (SFT + Verifier): Generating $3 \times 16$ completions using SFT generator to train a verifier, as described in §2.3. This is similar to ORM verification approach by (Cobbe et al., 2021) but empirically performs better (Fig. 5).
5. **V-STaR [1 Iter]**: Bootstrapping a generator and training a verifier for 1 iteration with $k = 3 \times 16$ completions from $G_{\text{SFT}}$, so that the total generation budget matches V-STaR. This baseline also corresponds to **RFT + Verifier**.
6. **Self-consistency / Majority Voting** (Wang et al., 2023a): An alternative to use test-time compute without verifiers: sample multiple solutions from STaR[†] generator and pick the majority-vote answer.

---

[2]This subset includes a total 150 problems of Level 1 difficulty in MATH with question types of *algebra*, *Counting & probability*, *prealgebra* and *number theory* where the final answer is a number and no latex exists in the question.

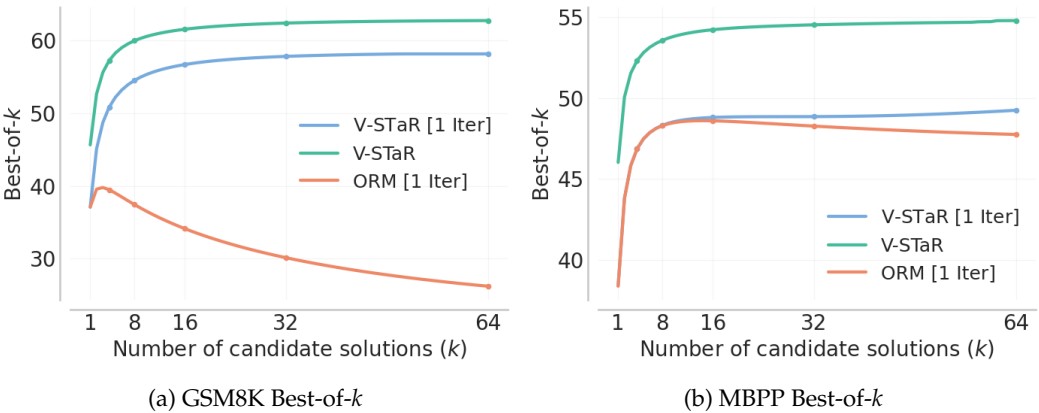

(a) GSM8K Best-of-$k$                                (b) MBPP Best-of-$k$

Figure 5: **Best-of-k test accuracy** of V-STaR, V-STaR [1 Iter], and outcome-supervised reward model (ORM) style verifier 7B models, measured by Eq. 3. Best-of-1 is equivalent to not having a verifier and is equal to Pass@1 of the generator.

At inference, we generate 128 candidate solutions for each test problem using the generator. We report Pass@1 accuracy for the generators, which estimates the probability that an answer randomly sampled from the generator is correct. Best-of-64 accuracy is used for all verifier-based methods, using the formula in (Eq. 3), as well as the self-consistency baseline.

## 4.2 Reliable estimation of Best-of-$k$ accuracy

To estimate accuracy with test-time verifiers, Cobbe et al. (2021); Lightman et al. (2024) repeat the following procedure several times and average the results: sample $k$ solutions, rank them using a verifier and take the top scoring one as the predicted answer. However, computing this best-of-$k$ accuracy can have high variance and is expensive. Instead, to measure the best-of-$k$ accuracy reliably, we propose two methods, akin to how Pass@k is computed (Chen et al., 2021). To do so, we estimate the probability that out of $k$ samples drawn without replacement from a fixed set of $N$ (for $N > k$) samples, the one with the highest verifier score is correct. The best-of-$k$ is calculated by repeating this process $M$ times and taking the average. Assuming there are no duplicate verifier scores this can be done efficiently (see §E in appendix), using the following formula:

$$\text{Best-of-}k := \frac{1}{\binom{N}{k}} \sum_{i=0}^{N-k} \binom{N-i-1}{k-1} \alpha_i \tag{3}$$

where $[\alpha_0, \ldots, \alpha_{N-1}]$ are the binary correctness values (0 or 1) for the $N$ candidates $y_i$ sorted in decreasing order by their verifier score. The numerator here can be derived by considering subsets where the top-ranked candidate is $y_i$ for all possible values of $i \in \{0, \ldots, N-k-1\}$.

## 4.3 V-STaR on Math Reasoning and Code Generation

As shown in Fig. 2, V-STaR shows consistent gains across GSM8K, MBPP, MATH subset and HumanEval test sets for LLaMA2 7B and 13B models (Fig. 8) over baselines. In math, we report absolute improvement of 6 to 17% in test accuracy over STaR[†] and Verification, and 4 to 12% in code generation tasks. The gains over V-STaR [1 iter] in Fig. 3 show that iteratively generating solutions to collect verifier training data results in a better distribution and quality compared to a non-iterative approach with the same generation budget. We show gains in each iteration of generator and verifier on MBPP in Fig. 7. We also trained a 4th iteration of generator and verifier on MBPP which led to a marginal gain of 0.3%.

**Out-of-domain performance of V-STaR**. The generators and verifiers trained on MBPP are evaluated on HumanEval, while those trained on GSM8K are evaluated on a subset of MATH test set (see Fig. 2 and Fig. 4). In general, we observe lower absolute Pass@1 and Best-of-64 scores for all methods as these two tasks are considered to be more difficult than GSM8K and MBPP. That said, Iterative V-STaR outperforms baselines, and V-STaR [1 iter] on both tasks and across model sizes. Utilizing incorrect solutions to train verifiers results in large improvements than just bootstrapping with correct model generated solutions using

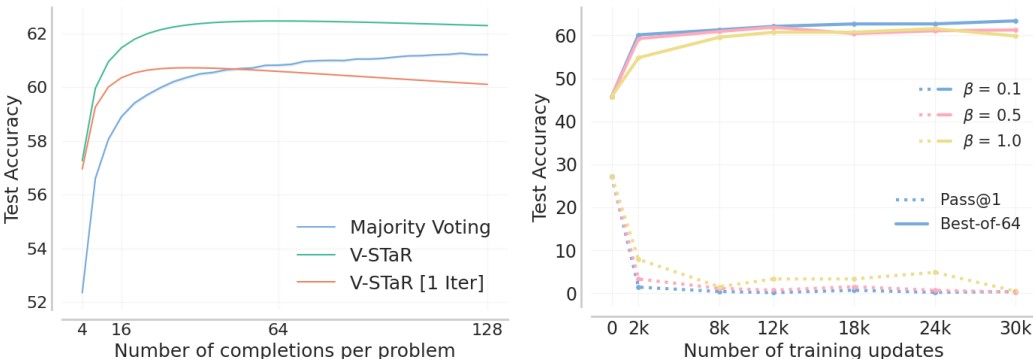

Figure 6: **Left: Best-of-k test accuracy** of 7B V-STaR compared to V-STaR[1 Iter] and self-consistency (Wang et al., 2023c) across different numbers of candidate solutions generated on GSM8K. We subsample solutions from $N = 1000$ generations. V-STaR can rank over a reasonably large number of candidate solutions. We compute 95% CIs for self-consistency using 256 trials. **Right: Comparing DPO-based generator and verifier** for V-STaR 7B, measured by Pass@1 and Best-of-64 respectively on GSM8K. Best-of-64 accuracy rises significantly while the generation ability of DPO verifier degrades with only 2k updates.

STaR$^{\dagger}$or RFT. While we use LoRA adapters due to compute constraints, we hypothesize that gains from V-STaR could potentially be larger with full parameter fine-tuning.

**Best-of-$k$ accuracy.** Fig. 5 shows test accuracy for $k = 1$ to $k = 64$, calculated from 128 candidate solutions per test problem, for 7B models on both tasks. Best-of-1 is equivalent to Pass@1 and ignores verifier scores. Best-of-$k$ saturates for $k \geq 16$ and the gap between V-STaR [1 Iter] and V-STaR stays consistent.

### 4.4 Comparing DPO *vs.* ORM verifiers

We trained ORM style verifiers, as described in §2.2, with LoRA adapters. These verifiers did seem to achieve relatively poor performance compared to DPO-based verifiers. Fig. 5a shows the comparison between the V-STaR [1 Iter] trained with DPO and an ORM style verifier on the same training data. ORM fails to effectively search through generated candidate solutions for number of candidates above 4 in the GSM8K task. The ORM style verifier is also performing worse than our DPO based verifier in MBPP for number of candidate solutions above 16.

### 4.5 How many completions can V-STaR be extended to?

Fig. 6 shows the Best-of-K accuracy of V-STaR 7B on GSM8K, measured by Eq. 3 as a function of $k$. V-STaR outperforms majority voting (Wang et al., 2023c) at searching over a large number of candidate solutions (see §F in appendix). While V-STaR is far more effective than majority voting for $k \leq 64$, the performance gap starts to slightly decrease for larger value of $k$, similar to performance decay reported in Cobbe et al. (2021). Furthermore, V-STaR can be used for any problem solving task where we can verify correctness while majority voting is **not** applicable to tasks such as code generation. We also tried combining verifier scores with reranking strategies, such as weighted reranking and weighted majority voting Liu et al. (2023), but did not observe performance gains.

### 4.6 Evaluating DPO verifier as a generator

Since DPO fine-tuned models can also be used as generators, we evaluate how good is the generation ability of DPO verifiers. Fig. 6 shows Pass@1 and Best-of-64 for V-STaR verifier over training updates, for three different $\beta$ coefficients for proximity to SFT policy in DPO objective (§3.1). The verifier's solving ability starts degrading only after a small number of training updates. In contrast, using the DPO objective for verification seems to be sample efficient as the model's Best-of-64 increases significantly with only 2$k$ training updates.

### 4.7 Should the verifier be in the training loop?

Optionally, one could train intermediate verifiers at each iteration and filter correct solutions to include in $\mathcal{D}_{\text{GEN}}$ and $\mathcal{D}_{\text{VER}}$ to provide feedback. This step seems more reasonable with sufficient exploration, that is larger values of $k$, when sampling $k$ solutions from the generator in each iteration.

We tried putting the verifier in the training loop to filter correct solutions from the generator for the next training iteration. To do so, we sampled $k = 64$ completions per query from the generator, labeled their correctness, and took only the top 8 based on their verifier score. We take as many samples from the incorrect set so that the total number of correct and incorrect completions per query is 16 or less. After running three iterations with verifier in the loop for MBPP, the final Best-of-64 accuracy and Pass@1 are 53.2 and 46.34 respectively.

Our results suggest that having the verifier in the training loop does not provide a substantial gain for this task. V-STaR is simpler without the verifier in the loop and there is no need to train a verifier at each iteration; however we did not experiment with other tasks and different sampling strategies from the generator at each iteration. We leave a more detailed study of this question to future work.

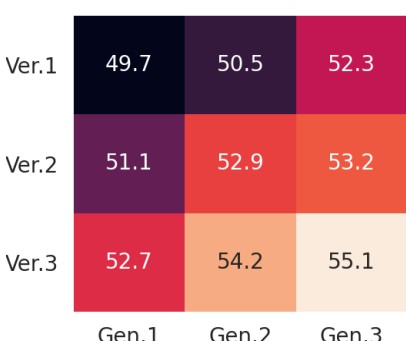

Figure 7: Best-of-64 for each V-STaR iteration on MBPP. The performance gets better with each iteration of generator and verifier without showing signs of collapsing.

### 4.8 Gains across V-STaR iteration

Fig. 7 shows the improvements achieved from each iteration of the generator and verifier on MBPP. The gains are larger across iterations of verifiers (from Ver.1 to Ver.3) than iterations of generators which highlights the importance of verifiers.

## 5 Related work

Challenging multi-step reasoning tasks has driven innovative research on LLMs, such as generating answers given questions via intermediate steps (Wei et al., 2022; Kojima et al., 2022). A large volume of recent work studies ways to improve the correctness of these intermediate steps and reducing the cost of arriving at a correct solution.

**Self-training and self-improvement.** One family of methods, beginning with STaR (Zelikman et al., 2022), reinforced self-training (Gulcehre et al., 2023), and rejection fine-tuning (Yuan et al., 2023), relies on solutions generated by the LLM to update itself. These methods fine-tune the model on generated solutions that yield a correct answer. ReST$^{EM}$ (Singh et al., 2023) view this fine-tuning as expectation-maximization based RL fine-tuning of a solution-generating agent. Wang et al. (2023a) propose a contrastive loss to make correct solutions more likely than incorrect ones, while Ni et al. (2023a) propose to use *intermediate* states of successful solutions as supervision to improve credit assignment. Discovery of successful solutions is a difficult exploration problem, and Luong et al. (2024) has shown that RL-based fine-tuning of a LLM is difficult unless it is initialized by some steps of supervised fine-tuning. In An et al. (2023), a more powerful LLM was used to edit the incorrect rationales generated by a smaller model and provide positive data for its fine-tuning. However, Huang et al. (2023) argued that LLMs are limited in their ability to correct their *own* reasoning. V-STaR is similar to self-improvement methods in that it uses its self-generated solutions for fine-tuning, but also trains a verifier using these solutions, including both correct and incorrect ones.

**Training verifiers.** Verifiers – models that score or rank reasoning chains with the aim of favouring successful rationales – were introduced for mathematical reasoning tasks by Cobbe et al. (2021), who proposed to collect correct and incorrect rationales from a tuned

generator and train a verifier. They noted the importance of a large training set for the success of the method. Uesato et al. (2022) found that process supervision – correctness of the rationale – enhances the performance of fine-tuned LLMs relative to outcome supervision – whether the answer is correct or not. Subsequent work correspondingly studied ways of deriving reward signals for individual reasoning steps (Li et al., 2023; Lightman et al., 2024; Yu et al., 2023), combining solution-level and step-level verifiers (Zhu et al., 2023), and augmenting verifiers with auxiliary information, such as results of program execution (Ni et al., 2023b). In Ma et al. (2023); Wang et al. (2023b), rationale generation is treated as a graph search problem, either using a stepwise verifier to guide the search or estimating the quality of steps by Monte Carlo rollouts. In V-STaR, the verifier is trained with DPO, which enjoys a high sample efficiency (see Fig. 6), and is used for ranking LLM-generated solutions at test-time.

The manner of training the verifier varies between works. The verifier can be viewed a reward model trained on human annotations – making the training of a generator that satisfies the verifier as an instance of RL with human feedback (Ziegler et al., 2019) – or on synthetic data, leading to forms of RL with AI feedback (Bai et al., 2022; Yang et al., 2023). The verifier can alternatively be viewed as a generative model, such as by conditioning on control tokens indicating a positive or negative label of a solution (Korbak et al., 2023) or by extracting the score as the likelihood of a special token following the candidate solution (Liu et al., 2023). V-STaR takes the unique approach of using DPO (Rafailov et al., 2023) to contrast the likelihoods of correct and incorrect solutions under the verifier (see §3.1).

## 6 Conclusion

We propose V-STaR, a data-efficient and simple to implement approach that utilizes correct and incorrect generated solutions from an iteratively trained generator to train a strong verifier. We find training verifiers with DPO to be more effective than the common method by Cobbe et al. (2021). Our empirical results show the effectiveness of V-STaR over existing self-improvement and verification-based methods. V-STaR has the potential to improve existing self-improvement loops on a wide range of problems with access to correctness feedback during training.

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

# A  Algorithm

---

**Algorithm 1:** V-STaR

---

**Input:** Original data $\mathcal{D}_{\text{SFT}}$, Training queries $\mathcal{D}_{\text{query}}$, base model $G_{\text{base}}$, Num generations $k$, Num iterations $T$

$\mathcal{D}_{\text{GEN}} \leftarrow \mathcal{D}_{\text{SFT}}$

$G_{\text{SFT}} \leftarrow \text{SFT}(G_{\text{base}}, \mathcal{D}_{\text{SFT}})$

**for** iter $= 1$ **to** $T$ **do**

  $G \leftarrow \text{SFT}(G_{\text{base}}, \mathcal{D}_{\text{GEN}})$          {fine-tune generator}

  $\mathcal{S} \leftarrow \text{sample}(G, \mathcal{D}_{\text{query}}, k)$         {generate candidates}

  $\mathcal{D}' \leftarrow \text{label\_correctness}(\mathcal{S})$       {score candidates to get $z$}

  $\mathcal{D}_{\text{GEN}} \leftarrow \mathcal{D}_{\text{GEN}} \cup \mathcal{D}'_{[z=1]}$      {correct solutions $\rightarrow$ buffer}

  $\mathcal{D}_{\text{VER}} \leftarrow \mathcal{D}_{\text{VER}} \cup \mathcal{D}'$       {all solutions $\rightarrow$ DPO buffer}

**end for**

$\mathcal{D}_{\text{pref}} \leftarrow \text{preference\_pairs}(\mathcal{D}_{\text{VER}})$

$V \leftarrow \text{DPO}(G_{\text{SFT}}, \mathcal{D}_{\text{pref}})$

---

# B  The prompt used for MBPP few-shot generation.

Following Ni et al. (2023b), we use the following prompt to sample completions per problem for data generation during training.

```python
# Write Python function to complete the task and pass the assertion tests.

### Task Start ###
# These are the assertions for your function:
assert similar_elements((3, 4, 5, 6),(5, 7, 4, 10)) == (4, 5)

""" Write a function to find the similar elements from the given two tuple lists. """
def similar_elements(test_tup1, test_tup2):
    res = tuple(set(test_tup1) & set(test_tup2))
    return (res)
### Task End ###

### Task Start ###
# These are the assertions for your function:
assert is_not_prime(2) == False

""" Write a python function to identify non-prime numbers. """
import math
def is_not_prime(n):
    result = False
    for i in range(2,int(math.sqrt(n)) + 1):
        if n % i == 0:
            result = True
    return result
### Task End ###

### Task Start ###
# These are the assertions for your function:
assert heap_queue_largest( [25, 35, 22, 85, 14, 65, 75, 22, 58],3)==[85, 75, 65]

""" Write a function to find the largest integers from a given list of numbers using
heap queue algorithm. """
import heapq as hq
def heap_queue_largest(nums,n):
    largest_nums = hq.nlargest(n, nums)
    return largest_nums
### Task End ###
```

## C  Test accuracy of 13B V-STaR and baselines

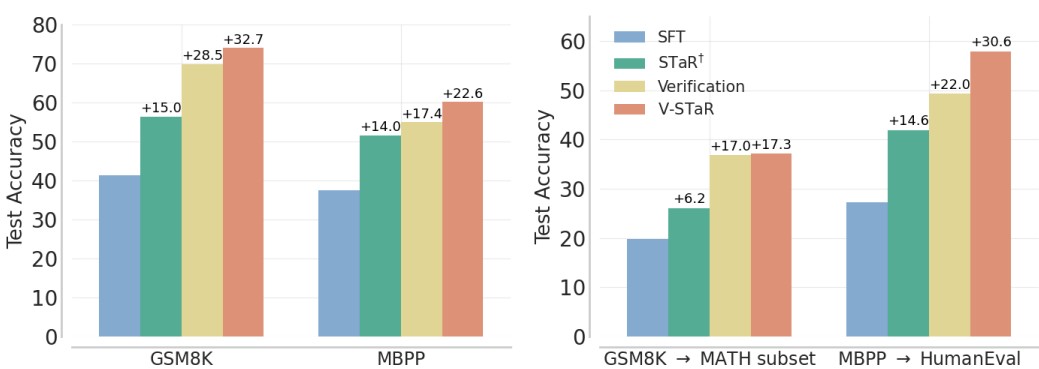

Figure 8: Test accuracy of 13B V-STaR compared to baselines. We report Best-of-64 for verification-based methods and Pass@1 for others. **(Left)** Test accuracy for training tasks. **(Right)** Transfer evaluation of GSM8K and MBPP trained models on MATH subset and HumanEval respectively.

## D  Candidate solutions $\hat{\mathbf{y}}_1$ and $\hat{\mathbf{y}}_2$ for a GSM8K problem $x$

**x** = Andy walks 50 meters from his house to school. After school, he comes back to the house and goes to the market. If he walks 140 meters in total, how many meters is the distance between the house and the market?

$\hat{\mathbf{y}}_1$ = He walks to school and back, which is 2 × 50 meters = 100 meters. So he walks 140 meters - 100 meters = 40 meters to the market. Answer=40

$\hat{\mathbf{y}}_2$ = If he walks 50 meters from his house to school, and 140 meters in total, he walks 140 - 50 = 90 meters from the school to the market. Answer=90

## E  Example for Best-of-$k$ using Eq. 3

*verifier scores*  [.9, .7, .3, .3, .2, .2, .1, 0, 0, 0]

$i$   0, 1, 2, 3, 4, 5, 6, 7, 8, 9

*sorted correctness $\alpha$*  [1, 0, 1, 1, 0, 1, 0, 1, 0, 0]

$$\text{Best-of-}4 = \frac{1 \times \binom{9}{3} + 0 \times \binom{8}{3} + 1 \times \binom{7}{3} + \cdots + 0 \times \binom{3}{3}}{\binom{10}{4}}$$

Figure 9: An example showing how Eq. 3 estimates Best-of-$k$ accuracy efficiently and without variance for $k = 4$ from $N = 10$ candidate solutions.

## F    V-STaR *vs.* majority voting example

**x** = Every day, Wendi feeds each of her chickens three cups of mixed chicken feed, containing seeds, mealworms and vegetables to help keep them healthy. She gives the chickens their feed in three separate meals. In the morning, she gives her flock of chickens 15 cups of feed. In the afternoon, she gives her chickens another 25 cups of feed. How many cups of feed does she need to give her chickens in the final meal of the day if the size of Wendi's flock is 20 chickens?

**V-STaR solution** = Wendi feeds each of her 20 chickens 3 cups of feed per day for a total of 20x3=60 cups of feed. She gives the chickens 15 cups in the morning and 25 cups in the afternoon for a total of 15+25=40 cups. Therefore, she needs to give the chickens 60-40=20 cups of feed in the final meal of the day. Answer=20

**Majority voting solution** = The total amount of feed Wendi gives the chickens is 15 + 25 = 40 cups of feed. Wendi has 20 chickens so she needs to give each chicken 40 / 20 = 2 cups of feed. Answer=2

## G    Best-of-$k$ using Eq. 3 *vs.* Best-of-$k$ from Lightman et al. (2024)

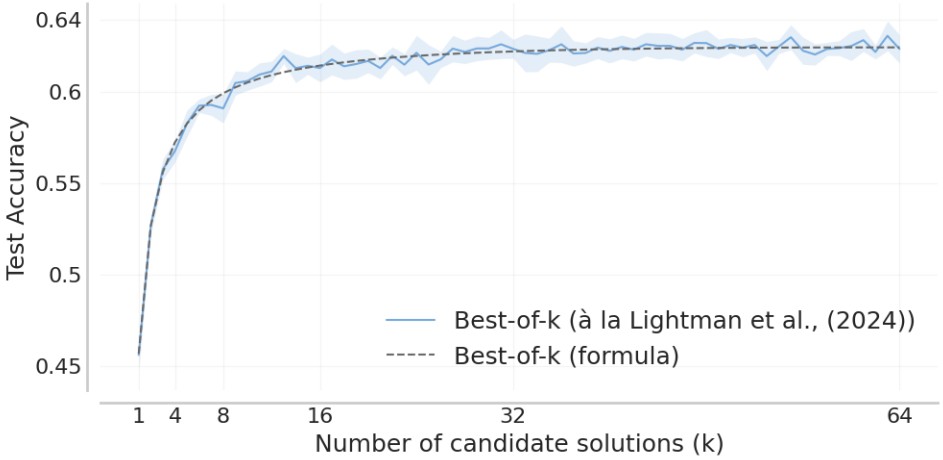

Figure 10: Eq. 3 estimates Best-of-$k$ accuracy efficiently and without variance. For Best-of-$K$ accuracy without the formula, the process is repeated 32 times.

