# OpenReview forum: "V-STaR: Training Verifiers for Self-Taught Reasoners"
_colmweb.org/COLM/2024/Conference — COLM_

### Official Review · Reviewer_H7mN · 2024-05-12

**Rating:** 7
**Confidence:** 4
**Ethics Flag:** 1

**Summary:**

This paper proposes V-STaR, an approach to boost small LMs' reasoning performance by combining two lines of existing work---1) Self-Taught Reasoner/Rejection Sampling Fine-tuning, where LMs are iteratively fine-tuned only on correct self-generated solutions and 2) Oversample-then-rerank, where multiple solutions are sampled for each test question and a verifier scores the correctness of each solution. Both in-domain and cross-domain evaluation results suggest the effectiveness of their approach across math reasoning and code  generation tasks. The intuition behind the improvement is that the verifier training learns from mistakes in the incorrect generations discarded by the STaR/RFT training.

**Reasons To Accept:**

1. A simple yet effective approach that combines the benefits of both STaR and verification to boost reasoning in fine-tuned smaller LMs.
2. The paper highlights the superiority of the DPO-based verifier over the previous ORM-based one, which is a novel discovery for me.
3. Improving reasoning in LLM is an important topic of widespread interest within the community. This paper will establish foundations for future work in this field.

**Reasons To Reject:**

1. While the improvements over the baseline are significant, the technical contribution is somewhat less exciting due to the straightforward combination of existing STaR and Verification techniques.

---

> ### Author Rebuttal · Authors · 2024-05-30
>
> Thank you for your thoughtful feedback.
>
> We are glad that the reviewer finds our approach simple, effective and improvements to be significant. As such, we believe that it will be straightforward for other people to adopt and extend our method.

---

### Official Review · Reviewer_opuK · 2024-05-14

**Rating:** 7
**Confidence:** 4
**Ethics Flag:** 1

**Summary:**

The paper presents an iterative approach to train LLMs to become better at math and code tasks by utilizing the models own generations. Concretely, the approach makes use of multiple samples generated by the model, keeping only the correct generations, and then training the model with this new subset. Additionally, the approach relies on a test time candidate selection model which is trained using the model generations using DPO, thereby further improving performance. The approach presents strong test accuracy improvement on popular math and code benchmarks.

**Questions To Authors:**

_Questions_

- Fig 6 is missing the 95% CI done over 256 trials - it would be good to add them here.
- Section 4.6 is confusing to me: why does the DPO trained verifier does poorly on generation? The model gets good at assigning log likelihoods to correct and incorrect trajectories, so wouldn't the generation reflect that too?
- Adding the verifier in training not getting substantial gain is a bit obvious, right? The verifier initially is noisy as it doesn't do a good job at filtering the correct solutions. Did you try a schedule of training verifier with manually filtered data from generation, and then using it during training?

_Suggestions_

- It would be interesting to measure the forgetting effect with more and more iterations. Does the model collapse when trained for increasing number of iterations?
- Does any linguistic pattern of the solutions emerge with increasing iterations?

**Reasons To Accept:**

- Simple method which works. Authors continually train generators by sampling trajectories and manually filtering the correct answers, thereby making the generator more tuned to output the correct answers.
- Combined with a verifier which is trained on DPO, the VSTAR approach makes good use of the verifier during testing to further prune likely incorrect solutions.
- Strong improvements on hard tasks such as GSM8k, MBPP (in-distribution) and MATH, HumanEval (out-of-distribution).

**Reasons To Reject:**

- It is not clear how many iterations ($T$) of training V-STAR is optimal. Authors use three iterations, but more study needs to be done on the impact of the number of iterations to find the ceiling of performance.
- Ablations are needed to understand how many iterations ($T$) of data collections are needed for the verifier as well. As the generator gets better at each iteration, the verifier should get harder data, thereby improving its performance.
- A small limitation of this paper is that the proposed approach is only verified against Llama 2 and CodeLLama. It is unclear whether this approach is general across several openly available models.

---

> ### Author Rebuttal · Authors · 2024-05-30
>
> **Summary:** Added transfer to Mistral, gains in each V-Star iteration, and addressed concerns
>
> We thank the reviewer for their insightful comments. We are glad that the reviewer likes our proposal of leveraging both positive and negative samples for self-improving the model. We are happy that the reviewer finds our improvements strong on hard in-distribution and out-of-distribution tasks.
>
> **Regarding training iterations:** We trained only for 3 iterations due to computation limits similar to Yuan et al., (2024). In [this](https://ibb.co/Y8rCsBY) plot we show the gains in each iteration for MBPP. We also trained a 4th iteration of generator and verifier for MBPP, and only observed a marginal improvement of 0.3% in Ver@64. The performance gets better with each iteration of generator and verifier without the models showing signs of collapsing. We will update the paper with this figure.
>
> **Other models:** Thank you so much for this suggestion. We tested our 7B Llama2-based verifier on candidate answers from Mistral 7B on GSM8K (8 shot). This verifier outperforms Majority-voting on Mistral 7B. This shows the transferability of our verifiers to other LLMs.
>
> Mistral 7B 8-shot:
> Pass@1: 32.2
> Maj@64: 58.6
> Ver@64: 60.2
>
> **Regarding Fig.6:** We do compute 95% CI in this figure as mentioned in the caption. The CI is extremely small.
>
> **DPO verifier as generator:** It is often the case that during training with DPO, the log-probability of preferred/correct samples goes down. This is observed in a number of works such as Rafailov et al., 2024. This could explain why our DPO trained verifier does poorly on generation.
>
> **Manually filtering data for verifier-in-the-loop:** Indeed this is a great suggestion. Experimenting with different sampling and filtering strategies at each iteration would be interesting and we hope that our work serves as the foundation for future work.
>
> **Emergence of linguistic patterns:** This is an interesting suggestion. We manually checked a few samples across iterations but did not identify any specific linguistic patterns.
>
>
> Yuan et al., 2024 Self-Rewarding Language Models
> Rafailov et el., 2024 : From r to Q∗: Your Language Model is Secretly a Q-Function.
>
>
> ___
>
> _We hope that your concerns/questions have been addressed. We’d be happy to engage in further discussions._

---

> > ### Comment · Reviewer_opuK · 2024-06-05
> > **Thanks for the response**
> >
> > Thanks for the detailed response and incorporating the suggestions. Adding the transferability results to the paper would be quite useful!

---

### Official Review · Reviewer_zjMQ · 2024-05-15

**Rating:** 7
**Confidence:** 4
**Ethics Flag:** 1

**Summary:**

The paper modifies STaR (Zelikman et al., 2022) to make use of the sampled incorrect solutions by training an additional verifier that is used at test time to rank candidate solutions. To make use of both correct and incorrect solutions, the authors propose to train the verifier as a reward model using Direct Preference Optimization. During each iteration of STaR, the verifier is trained on a union of the collected samples so far from the policy model.

**Questions To Authors:**

* How was the ORM trained? and what is the intuition behind why it performs very poorly compared to DPO? Khalifa et al., 2023 mentions that ORM can perform poorly if many of the correct solutions involve bad reasoning and the LM was able to reach these solutions arbitrarily. Do you have a similar observation? It would be interesting to draw connection with existing work.
* Why train only for 3 iterations? What happens to the verifier performance if you train for more iterations? I'm surprised you do not have a plot showing verifier performance after each STaR iteration.

**References:**

Khalifa et al., "Discriminator-guided multi-step reasoning with language models." arXiv preprint arXiv:2305.14934 (2023).

**Reasons To Accept:**

* Paper is well written with a clear motivation and contribution
* Impressive performance boost on different reasoning tasks.
* The direction of leveraging preference optimization approaches for learning from suboptimal solutions is interesting and exciting.

**Reasons To Reject:**

* Why use DPO? I feel the justification for using DPO is not clear, and no analysis is done to understand why it is performing better than standard ORM.
* Lack of analysis on the trained verifier. It would be very interesting to see if the trained verifier was able to learn a process reward model (PRM) by only training using final answer labels.

---

> ### Author Rebuttal · Authors · 2024-05-30
>
> **Summary:** Figure showing verifier performance for each iteration, justifying DPO, answered questions
>
> Thank you for your valuable feedback. We are glad that the reviewer finds our paper well written with a clear motivation and contribution, and finds our proposed method of leveraging preference optimization to learn from correct and incorrect solutions interesting.
>
> **Training iterations:** We trained only for 3 iterations due to computation limits similar to Yuan et al., (2024). In [this](https://ibb.co/Y8rCsBY) plot we show the gains in each iteration for MBPP. We also trained a 4th iteration of generator and verifier for MBPP, and only observed a marginal improvement of 0.3% in Ver@64. We will update the paper with this figure.
>
> **Why DPO:** The classification loss and the language modeling loss of typical verifiers could be joined as the DPO objective. The task of learning a verifier seems like a reward modeling task, and we made this connection to train with DPO. We found DPO trained verifiers to be more robust than ORM style verifiers for math and coding problems (Figure 5 in the paper). This could be partly due to using LoRA adaptors or the contrastive nature of DPO’s objective. Due to compute constraints, we are not able to test DPO verifiers against ORMs (à la Lightman et al., 2024) without LoRA adapters. Coincidentally, some of the top reward models on Reward Bench (Lambert et al., 2024) are also DPO based.
>
> **ORM as PRM:** Since our verifier outputs a log-probability for a candidate answer, we can only use them as ORMs to rank candidate solutions. That said, we manually inspected a number of model generated solutions under the verifier to see if the log-probability starts to drop at a certain step for wrong solutions, but this was not the case.
>
> **ORM details and false positives:** We train our ORMs using a combination of language modeling and a classification objective as discussed in sec. 2.2. Our ORMs are trained with the same data as our DPO based verifiers. It is possible that DPO is more robust against false positives since DPO’s contrastive objective only assumes that a false positive is **better** than an incorrect response, but ORM assumes that a false positive response **is correct**.
>
> Lightman et al., 2024 Let's Verify Step by Step
> Lambert et al., 2024 RewardBench
> Yuan et al., 2024 Self-Rewarding Language Models
>  ___
> _We hope that your concerns/questions have been addressed. We’d be happy to engage in further discussions._

---

> > ### Comment · Reviewer_zjMQ · 2024-06-05
> >
> > Thank you for the rebuttal and the clarifications.
> >
> > I'm still not convinced as to why DPO was chosen to train the verifier as opposed to a simple contrastive objective such as $\log({\sigma(r(y^+)) - \sigma(r(y^-))})$ as was used by Khalifa et al, and others. My intuition is that this will perform just as well as DPO if not better. I believe this is a crucial baseline.

---

> > ### Author Response · Authors · 2024-06-05
> >
> > Thank you for your further comments.
> >
> > Our main contribution is training verifiers using negative data generated during the self improvement process. As such, how we train verifiers is an empirical choice and depends on several factors such as using full or parameter efficient fine-tuning. Nevertheless, we compared two methods of training verifiers: (1) ORM (Cobbe et al, 2021) which corresponds to reward modeling combined with SFT, (2) DPO. Empirically, we found that when using LoRA, DPO results in better verifiers for GSM8K and comparable on MBPP, as shown in Figure 5.
> >
> > That said, V-STaR can be instantiated with any approach for training verifiers, and we have launched verifier training with contrastive loss used by Khalifa et al, as we believe it would make our work stronger! We’d include these results in the revision and post here as well if the experiments finish in time.

---

### Official Review · Reviewer_THgs · 2024-05-16

**Rating:** 6
**Confidence:** 4
**Ethics Flag:** 1

**Summary:**

The paper introduces a method that leverages the incorrect reasoning paths generated by LLM during SFT progression to train a Verifier. This Verifier, in turn, helps in selecting the correct answer during inference. The experimental results demonstrate a significant improvement in performance on both the math dataset and the code generation dataset, and the model also exhibits strong performance in transfer experiments.

**Questions To Authors:**

Please refer to questions.

**Reasons To Accept:**

1. This paper pioneers the utilization of incorrect labels to train a verifier, showcasing its potential utility across various scenarios.
2. The experimental results reveal a significant improvement in performance for Llama2 across both the math and code datasets.

**Reasons To Reject:**

1. The effectiveness of the verifier remains underexplored. While the paper compares it with majority voting in their baseline, further investigation is warranted given the crucial role the verifier plays. For instance, it is imperative to determine whether the verifier outperforms self-evaluation when selecting answers [1] and during decoding [2] [3]. Failure to do so may leave the utility of the verifier incompletely assessed.
2. The transferability of the verifier remains unexplored. Demonstrating its ability to be transferred to other models would be valuable, as it would reduce the need to train independent verifiers for each model, thereby streamlining the process.
3. The method is tested on Llama2 whose reasoning ability is far from the sota model such as mistral and gemma, which may make the method less general.

[1] Adaptation with Self-Evaluation to Improve Selective Prediction in LLMs, EMNLP 2023

[2] Tree of Thoughts: Deliberate Problem Solving with Large Language Models, Neurips 2023

[3] Self-Evaluation Guided Beam Search for Reasoning, Neurips 2023

---

> ### Author Rebuttal · Authors · 2024-05-30
>
> **Summary:** Added self-eval comparison, transfer to Mistral and addressed concerns
>
> Thank you for your comments. We are glad that the reviewer finds our method pioneering the utilization of negative data and our improvement significant.
>
> **Comparison with self-evaluation:** Based on your suggestion, we tried using the last generator (iteration 3) of GSM8k as the verifier to pick the best answer using the log-probability of the answer under the generator. This self-evaluation method outperforms pass@1 but substantially underperforms majority-voting and V-STaR.
>
> Pass@1: 45.6
> Majority@64: 60.9
> Self-evaluation@64: 50.9
> V-STaR@64 : 62.7
>
> **Transferability of verifier:** To do so, we tested our 7B Llama2-based verifier on candidate answers from Mistral 7B on GSM8K (8 shot). This verifier outperforms Majority-voting on Mistral 7B. This shows the transferability of our verifiers to other LLMs
>
> Mistral 7B 8-shot:
> Pass@1: 32.2
> Maj@64: 58.6
> Ver@64: 60.2
>
> **Models:** As mentioned earlier, our preliminary results on Mistral 7B show the transferability of our verifiers. For clarification, Mistral 7B Maj@64 is well below that of our iteration 3 generator based on LLama2 7B. All of the baselines to which we compare V-Star to are finetuned LLama 2 or CodeLlama models.
> Experiments on Gemma could not have been done as it was released too late (Feb 21st), and we did not have the time or computation to experiment with it.
> ___
>
> _We hope that most of your concerns have been addressed and, if so, we would appreciate it if you could reconsider your assessment. We’d be happy to engage in further discussions._

---

> > ### Comment · Reviewer_THgs · 2024-06-06
> > **Reply to Rebuttal**
> >
> > Thank you for incorporating additional experiments, which partially address my concern. Taking into account the transferability of the verifier, which is beneficial for the scalability and applicability of the method, I have revised my score to 6. However, the self-evaluation experiments are not convincing. My initial thought was to not use the verifier that you trained at all, but to use the self-evaluation of the model itself as the verifier.

---

> ### Author Response · Authors · 2024-06-05
>
> Dear Reviewer THgs,
>
> We would like to thank you again for your detailed reviews.
>
> In short, we added comparison with self-evaluation based on your suggestion, transferability of verifier to Mistral 7B on GSM8K, and addressed your concerns about using other models.
>
> We would be happy to do any follow-up discussion or address any additional comments.

---

> ### Author Response · Authors · 2024-06-06
> **Re-self evaluation results**
>
> There might be some confusion about our self-evaluation results.
>
> As mentioned by the reviewer, we ignored the verifier indeed and used the self-evaluation of the model (which we call generator), using the log probability of generated solution under the model, to score 64 solutions and pick the one with highest self-evaluation score. We found that doing so (self-eval@64) was better than pass@1 performance but worse than using the verifier (verifier@64) to pick the solution. Here, our generator correspond to running STaR for 3 iterations (and does not involve the verifier).
>
> Hope this clarifies the results we reported in the rebuttal. Please let us know if you were thinking something else for the self-evaluation baseline.

---

> > ### Comment · Reviewer_THgs · 2024-06-06
> > **Reply to Authors**
> >
> > Thank you for the further clarification and explanation. This addresses some of my concerns about the self-evaluation experiments. I would probably still suggest including experiments comparing to other standard baselines (like Tree-of-Thought or Self-Evaluation Beam Search) in the next version.

---

### Author Response · Authors · 2024-06-05

We are grateful for all of the reviewers' constructive and valuable comments. The reviewers consider V-STaR to be simple, effective, and foundational for future work in this field. We are happy that they find our improvements impressive and strong on hard math and coding tasks. Based on reviewers’ feedback, we have made following main changes:
* **Comparison with self-evaluation** showing that it substantially underperforms majority-voting and V-STaR.
* We will add [this plot ](https://ibb.co/Y8rCsBY) **showing gains in each V-STaR iteration** for MBPP.
* We also trained a 4th iteration of generator and verifier for MBPP which led to a marginal improvement of 0.3% in Ver@64.
* **Transferability of the verifier** by testing our 7B LLama2-based verifier on candidate answers from Mistral 7B on GSM8k. This verifier outperforms Majority-voting on Mistral 7B.

---

### Decision · Program_Chairs · 2024-07-10

**Decision:**

Accept

**Comment:**

This paper introduces V-STaR to enhance the reasoning capabilities of small LMs on math and code reasoning tasks by leveraging their own generations. V-STaR integrates two main strategies: 1) Self-Taught Reasoner/Rejection Sampling Fine-tuning, where LMs are iteratively fine-tuned on correct self-generated solutions, and 2) Oversample-then-rerank, where multiple solutions are generated and a verifier scores each solution to identify the most accurate one. This verifier is trained using a combination of correct and incorrect solutions to discern the quality of generated outputs. Experimental results demonstrate significant improvements in performance on both in-domain and cross-domain tasks.

Overall, this paper is novel in terms of using incorrect responses in training a verifier, which shows substantial improvements in Llama2 performance. The approach is simple yet effective, with the results demonstrating significant performance boosts on challenging tasks and showing promise in boosting reasoning capabilities in smaller fine-tuned LMs. There are several places the work can improve:

- The effectiveness and capabilities of the verifier can be further explored.

- Moreover, analysis of why DPO is preferred over ORM and whether the verifier can function effectively as a process reward model is needed to justify the contribution of the proposed method.

- Further experiments are needed to apply the methods beyond Llama2.